# Benchmarking a highly selective USP30 inhibitor for enhancement of mitophagy and pexophagy

Emma V Rusilowicz-Jones[1],*, Francesco G Barone[1],* , Fernanda Martins Lopes[2], Elizabeth Stephen[2], Heather Mortiboys[2] , Sylvie Urbé[1], Michael J Clague[1]

The deubiquitylase USP30 is an actionable target considered for treatment of conditions associated with defects in the PINK1-PRKN pathway leading to mitophagy. We provide a detailed cell biological characterization of a benzosulphonamide molecule, compound 39, that has previously been reported to inhibit USP30 in an in vitro enzymatic assay. The current compound offers increased selectivity over previously described inhibitors. It enhances mitophagy and generates a signature response for USP30 inhibition after mitochondrial depolarization. This includes enhancement of TOMM20 and SYNJ2BP ubiquitylation and phosphoubiquitin accumulation, alongside increased mitophagy. In dopaminergic neurons, generated from Parkinson disease patients carrying loss of function PRKN mutations, compound 39 could significantly restore mitophagy to a level approaching control values. USP30 is located on both mitochondria and peroxisomes and has also been linked to the PINK1-independent pexophagy pathway. Using a fluorescence reporter of pexophagy expressed in U2OS cells, we observe increased pexophagy upon application of compound 39 that recapitulates the previously described effect for USP30 depletion. This provides the first pharmacological intervention with a synthetic molecule to enhance peroxisome turnover.

## Introduction

The ubiquitin-specific peptidase (USP) family of proteins represents an emerging focus for drug discovery efforts (1). Amongst this family, USP30 is unique in exclusively localizing to mitochondria and peroxisomes, by virtue of its transmembrane domain (2, 3, 4). At mitochondria, it is optimally positioned to oppose the PINK1-PRKN–mediated cascade of ubiquitylation that follows mitochondrial depolarization and leads to mitophagy (5, 6). Thus, inhibition of USP30 represents an actionable target to correct pathologies associated with PINK1 or PRKN defects, such as Parkinson disease

or pulmonary fibrosis (7, 8). We and others have shown that USP30 localizes to peroxisomes and that its depletion can lead to elevation of pexophagy without any effect on bulk macroautophagy/autophagy (4, 9). Peroxisome levels determine the abundance of ether-linked lipids (plasmalogens) which are important for ferroptotic cell death in multiple cancer cells and are reduced in the brains of Alzheimer patients (10, 11). Current tools to manipulate peroxisome turnover are limited to manipulation of lipids in the specific context of hepatocytes or crude approaches that stimulate general autophagy (12, 13, 14).

Recent studies have reported highly related cyanopyrrolidine covalent USP30 inhibitors that recapitulate signature effects of USP30 depletion in the context of acute mitochondrial depolarization (15, 16, 17). These include enhanced TOMM20 ubiquitylation, increased mitophagy and greater accumulation of pUb (phosphoSer65 ubiquitin) (6, 15, 16, 17, 18). In the case of FT385, the specificity of the drug across a panel of DUBs (deubiquitylases) was high at concentrations up to 200 nM but it becomes less specific at higher concentrations. In the optimal concentration range, it is recommended to define target engagement for each cell line under study using a competition assay with reactive ubiquitin probes (16, 19). Phu et al. used higher concentrations of a related compound in a large scale proteomic analysis. By comparison with $USP30^{-/-}$ cells, they were able to characterize both on and off target effects of the drug (15).

In light of these limitations, we decided to characterize a benzosulphonamide, from a series of compounds, described in the literature as specific USP30 inhibitors. Compound 39 (CMPD-39) has a reported IC50 of ~20 nM in an in vitro assay of enzyme activity. In the same study, a related compound was shown to accelerate the turnover of mitochondrial DNA in terminally differentiated C2C12 cells, but otherwise their biological effects are uncharacterized (20, 21). Using USP30 KO cells for comparison and off target assessment, our results validate a highly selective inhibition of USP30, which recapitulates the signature associated with structurally unrelated inhibitors. Moreover, we show for the first time that chemical inhibition of USP30 can increase basal pexophagy to an extent consistent with previous observations of USP30 deletion and depletion (4, 9).

[1]Department of Molecular Physiology and Cell Signaling, Institute of Systems, Molecular and Integrative Biology, University of Liverpool, Liverpool, UK    [2]Sheffield Institute for Translational Neuroscience (SITraN), University of Sheffield, Sheffield, UK

Correspondence: urbe@liv.ac.uk; clague@liv.ac.uk
*Emma V Rusilowicz-Jones and Francesco G Barone contributed equally to this work

## Results

Previous work has reported IC50 values for a series of benzosulphonamide inhibitors, of which all those <1 $\mu$M were screened for selectivity, in comparison to three other USP family members (20, 21). We have selected one of these compounds (CMPD-39, Fig 1A) for a more rigorous test of selectivity provided by the Ubiquigent DUB profiler screening platform comprising >40 DUBs. CMPD-39 is a highly selective inhibitor of USP30 over two orders of magnitude of concentration (1-100 $\mu$M, Fig 1B). To test engagement of the inhibitor with USP30 applied to intact cells, we assayed competition with Ub-PA (Ub-propargylamide). Ub-PA covalently binds to the USP30 active site leading to a characteristic upshift on a SDS–PAGE gel (22). If a drug blocks access to the catalytic site, then the probe modification is blocked and the apparent molecular weight of USP30 shifts down accordingly. Applying CMPD-39 to SHSY5Y neuroblastoma cells shows strong competition for Ub-PA in the sub $\mu$M range of concentrations (Fig 1C). A robust proxy read-out for target engagement is the enhancement of TOMM20 ubiquitylation after mitochondrial depolarization (6, 16). This second assay indicates a maximal effect with 200 nM CMPD-39 applied to RPE1-YFP-PRKN cells (Fig 1D).

We next took advantage of our previously described isogenic USP30 KO RPE1-YFP-PRKN and SHSY5Y cell lines to demonstrate the target dependence of four signature effects of USP30 inhibition. In RPE1-YFP-PRKN cells, 4 h of mitochondrial depolarization is sufficient to observe a substantive loss of TOMM20 due to mitophagy. This can be enhanced in cells treated with CMPD-39 but not in the isogenic USP30 KO cells (Fig 2A). Depolarization induced TOMM20 ubiquitylation is enhanced by CMPD-39 after only 1 h in YFP-PRKN over-expressing RPE1 cells and after 4 h in SHSY5Y cells expressing endogenous PRKN. However, in USP30 KO cells of both types, the signal is already maximally elevated and is impervious to the inhibitor (Fig 2B, C, E, and F). The same pattern holds for a second biomarker, SYNJ2BP (synaptojanin 2–binding protein), identified by global ubiquitomic analysis of the response to FT385, a cyanopyrrolidine class of USP30 inhibitor (Fig 2B, D, and E) (16). Mitochondrial depolarization leads to the accumulation of the kinase PINK1, which phosphorylates ubiquitin (pUb) and PRKN at Ser65 (23, 24). This serves to recruit and activate PRKN, setting off a cascade of ubiquitylation at the mitochondrial surface (25). USP30 inhibition with cyanopyrrolidine inhibitors has the effect of modestly enhancing the PINK1 dependent accumulation of pUb on mitochondria (16, 17). In SHSY5Y cells this is most apparent between the 38 and 76 kD molecular weight range and is now reproduced with CMPD-39 or KO of USP30 with no additive effect (Fig 2E and G).

So far, the biochemical signatures we have presented require acute mitochondrial depolarization, which represents a non-physiological condition. Under basal conditions only a small fraction of mitochondria is undergoing mitophagy. We have quantitated this fraction using SHSY5Y cells stably expressing the fluorescent mitophagy reporter mitoQC (mCherry-GFP-Fis1(101-152)) (26). At 1 $\mu$M CMPD-39, the number and size of mitolysosomes is increased, in line with previous observations using FT385 (Fig 3A–C) (16). Interest in the therapeutic potential of USP30 inhibitors has been driven by their application in Parkinson disease,

for which some patients have loss of function mutations in the PRKN gene. We next turned to dopaminergic iNeurons (induced neurons) generated from patient derived fibroblasts, via iNPCs (induced neuronal progenitor cells) (27). In this experiment we quantitate the co-localization of lysotracker with the mitochondrial marker TMRM (tetramethylrhodamine) as an indicator of mitophagy in conjunction with the Opera Phenix screening platform, as previously described (27). The level of mitophagy in neurons from two PRKN compound heterozygotic mutant patients was reduced compared with two controls, in line with previous observations (27). Upon application of CMPD-39 to the neuronal cultures for 24 h, the control samples showed a trend towards increased mitophagy. Most notably, both PRKN mutant cell lines showed a statistically significant increase, such that their mitophagy index was restored to near control levels (Fig 3D). No toxic effects of CMPD-39 on neurons were observable over 96 h incubation periods, even at the highest concentrations (1 $\mu$M). It could be possible that CMPD-39 enhances mitophagy by dissipating the mitochondrial membrane potential. We did not observe a disruption of membrane potential at any of the concentrations used. In fact, treatment of CMPD-39 was able to correct deficits in mitochondrial membrane potential in PRKN mutant samples, which have a lower baseline potential as previously reported (Fig 3E) (27).

Finally we examined the effect on basal pexophagy, which is a PINK1 and PRKN independent pathway, but can nevertheless be enhanced by USP30 depletion without a corresponding gain in non-selective autophagy (4, 9). For this, we used U2OS cells stably transfected with Keima-SKL, a fluorescence reporter for pexophagy (4). When peroxisomes reach the acidic lysosome compartment, the excitation spectrum of the Keima fluorophore undergoes a shift that is then pseudo-colored as red in the illustrative images. CMPD-39 induced a strong increase in basal pexophagy (Fig 4), in line with our previous observations, obtained by USP30 KO or depletion with specific siRNAs (4). To our knowledge, this is the first example of a synthetic compound with a clearly defined target that can promote pexophagy.

## Discussion

Across the neurodegeneration landscape, actionable targets are still relatively scarce. Much attention has focused on enhancing mitophagy. In PD, this has been driven by the understanding that two PD-associated genes PINK1 and PRKN combine to promote mitophagy and that this pathway is deficient in patients bearing loss-of-function mutations (28). Defects in mitochondria are also evident in idiopathic patients or those with mutations in other PD-associated genes such as LRRK2 (leucine-rich repeat kinase 2) (29, 30, 31). Furthermore, mitophagy is impaired in Alzheimer disease (32, 33), and loss-of-function mutations in the mitophagy regulator TBK1 (TANK-binding kinase 1) and the receptor protein, optineurin, are associated with ALS (Amyotrophic Lateral Sclerosis) (34).

The PINK1/PRKN pathway is the best understood mitophagy pathway and is propelled by a PRKN-mediated cascade of ubiquitylation at the mitochondrial surface. As the most prominent mitochondrial deubiquitylase, USP30 is ideally placed to limit this

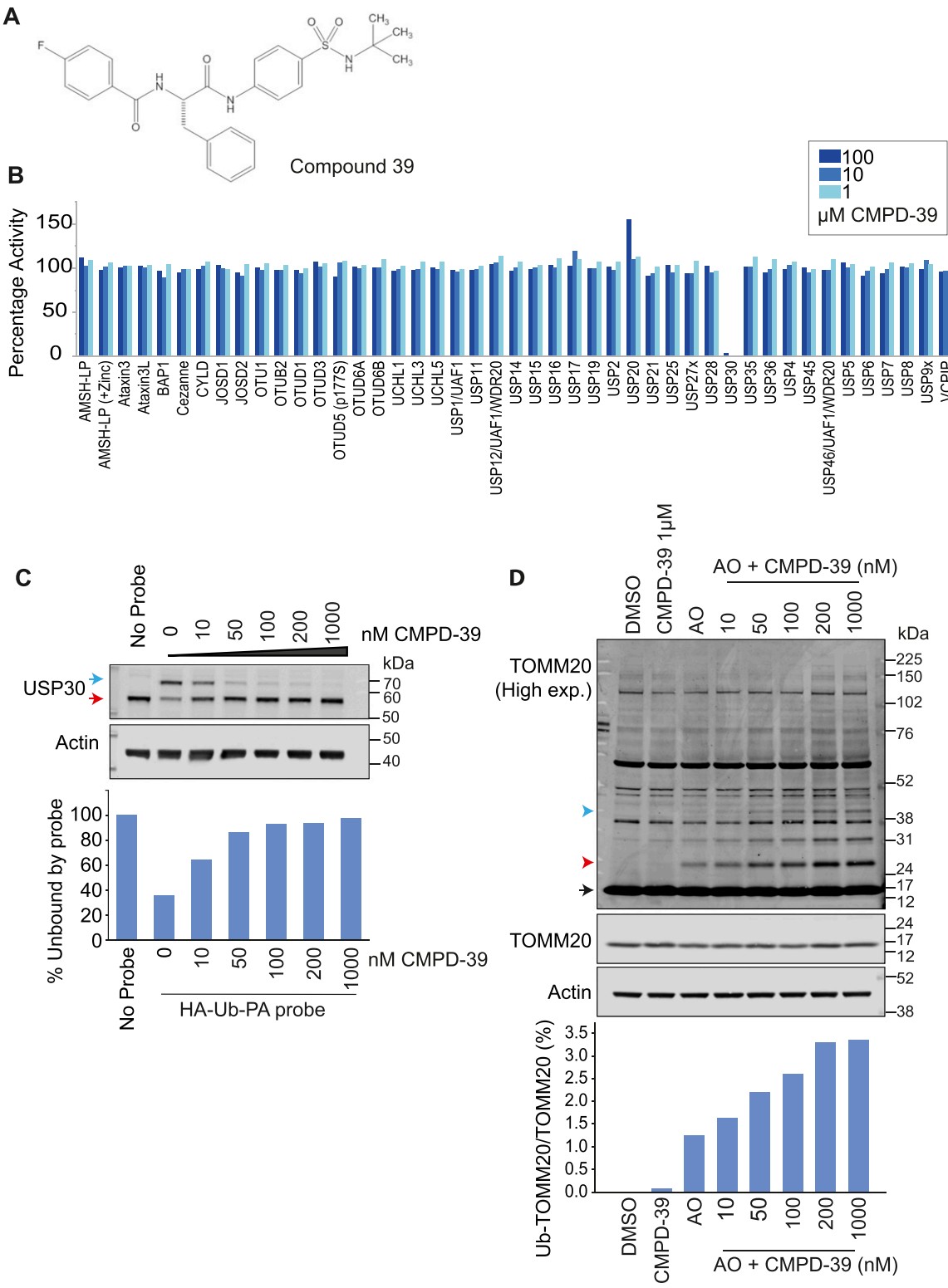

**Figure 1.  CMPD-39 is a selective USP30 inhibitor.**
**(A)** Chemical structure of CMPD-39. **(B)** DUB specificity screen (DUB profiler, Ubiquigent) with 1-100 μM CMPD-39. **(C)** Activity-based ubiquitin probe assay shows that CMPD-39 engages USP30 in cells at nanomolar concentrations in intact SHSY5Y cells. Samples were incubated with CMPD-39 for 2 h at the indicated concentrations, then incubated with HA-Ub-PA probe for 10 min at 37° C and immunoblotted as shown. Red arrow indicates unbound USP30; blue arrow represents probe bound USP30. **(D)** Inhibition of USP30 enhances the ubiquitylation of TOMM20 in YFP-PRKN over-expressing hTERT-RPE1 cells in a concentration dependent manner in response to mitophagy induction. Cells were treated for 1 h with DMSO or antimycin A and oligomycin A (AO; 1 μM each) in the absence or presence of CMPD-39 at the indicated concentrations, lysed, and analyzed by Western blotting. Black arrow indicates unmodified TOMM20, ubiquitylated species are indicated by red (mono-ubiquitylated) or blue arrow heads. Quantitation shows percentage mono-ubiquitylated.
Source data are available for this figure.

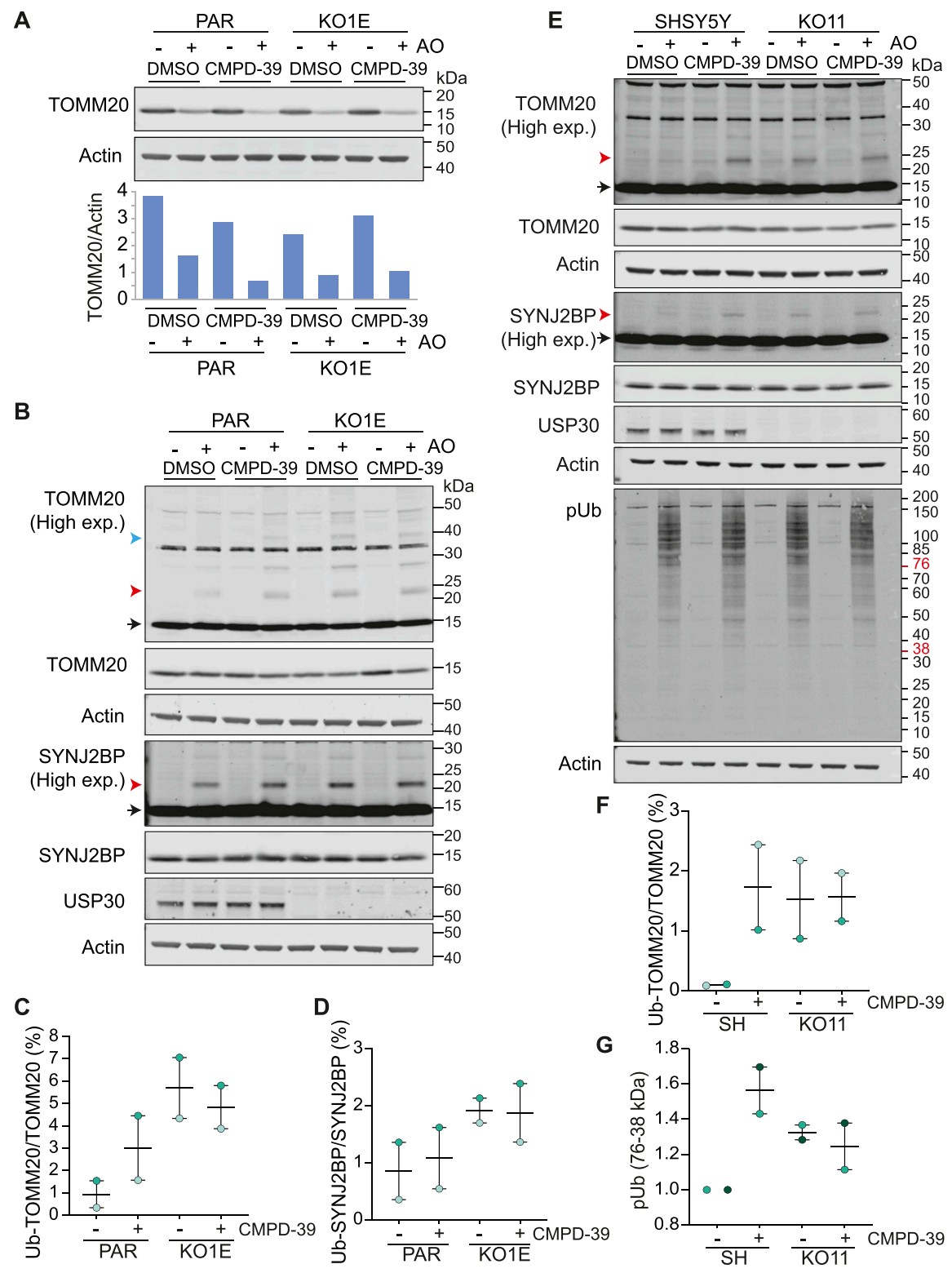

**Figure 2. CMPD-39 promotes depolarization-dependent ubiquitylation of previously described USP30 substrates.**
**(A)** USP30 inhibitor (CMPD-39) treatment of Parental (PAR) hTERT-RPE1-YFP-PRKN cells phenocopies *USP30* deletion (KO1E) by promoting TOMM20 degradation, whereas TOMM20 degradation is unaffected by CMPD-39 in the *USP30* KO (KO1E) cells. Cells were treated for 4 h ± AO (1 µM) ± 200 nM CMPD-39. **(B)** TOMM20 and SYNJ2BP ubiquitylation is enhanced within 1 h by CMPD-39 but is unaffected in the *USP30* KO (KO1E) cells. Cells were treated for 1 h ± AO (1 µM) ± 200 nM CMPD-39. **(C, D)** Graphs show quantification of mono-ubiquitylated TOMM20 and SYNJ2BP in AO-treated samples as a percentage of total for two independent experiments (error bars indicate the range). **(E)** USP30 inhibitor (CMPD-39) treatment of cells expressing endogenous PRKN (SHSY5Y cells) similarly phenocopies *USP30* deletion (KO11) by promoting TOMM20

pathway (3). If that is the case, then chemically inhibiting USP30 offers a therapeutic opportunity to enhance mitophagy. A further attractive feature of USP30 inhibition is that it is a nonessential gene, and this is a necessary condition for any long-term treatment option. The best characterized USP30 inhibitors to date are a series of cyanopyrrolidines, for which a consensus biochemical and cell physiological signature is beginning to emerge (15, 16, 17). One limitation of the use of these compounds is that they become less specific at higher concentrations and this can become problematic if the cellular uptake and target engagement efficiency is unknown (15, 17). We wondered if the recently published series of benzosulphonamide USP30 inhibitors might represent a structurally independent alternative, offering a greater dynamic range of concentration over which inhibition is achieved and specificity is retained (20). The cell physiological characterization of these compounds has hitherto been rudimentary. We now show that CMPD-39 is a highly selective USP30 inhibitor, which can achieve maximal USP30 inhibition at <200 nM in SHSY5Y cells but retains selectivity amongst DUB family members even up to 100 $\mu$M. Our data obtained with this molecule suggest that it is likely to provide a more robust tool compound than the cyanopyrrolidines. We reinforce the consensus signature for USP30 inhibition, that is, (i) increased TOMM20 and SYNJ2BP ubiquitylation following mitochondrial depolarization, (ii) moderately enhanced pUb accumulation following mitochondrial depolarization, and (iii) enhancement of basal mitophagy.

It is known that dopaminergic neurons derived from patients carrying loss of function *PRKN* mutations show lower levels of mitophagy (27). Here, we now show that this defect can be restored by a specific inhibitor of USP30, providing further encouragement for preclinical development of these compounds. Some overlapping results have recently been reported for a set of cyanopyrrolidine inhibitors (17).

Peroxisomes are strongly linked to neurodegenerative disease (35, 36). They support oxidation of various fatty acids and regulate redox conditions. They are involved in synthesis of ether linked phospholipids, notably plasmalogens, that are highly enriched in the nervous system. A fraction of USP30 associates with peroxisomes and its loss leads to an increase in pexophagy (4, 9, 37). siRNA depletion or KO of USP30 in RPE1 cells enhances pexophagy and importantly this effect can be rescued by re-expression of USP30, but not a catalytically inactive mutant (4). We now show that chemical inhibition of USP30 also enhances pexophagy in U2OS cells. The ability to acutely manipulate this process in a specific manner has been lacking and we anticipate that CMPD-39 could become a widely adopted tool compound for this alternative application. In summary, benzosulphonamide USP30 inhibitors, specifically CMPD-39, offer an important new class of tool compound for enhancement of mitophagy and pexophagy.

# Materials and Methods

## Cell culture

hTERT-RPE1-YFP-PRKN, SHSY5Y, SHSY5Y-mitoQC (mCherry-GFP-Fis1(101-152)), and U2OS-Keima-SKL were routinely cultured in DMEM/F12 (31331028; Gibco) supplemented with 10% FBS (10270106; Gibco) and 1% nonessential amino acids (11150035; Gibco). USP30 KO cells were generated as described in Rusilowicz-Jones et al (16). Primary fibroblasts were obtained from Coriell Cell Repository ([coriell.org](http://coriell.org)); control A GM13335 (M57), control B ND29510 (F55), *PRKN* A ND30171 (M54, Arg42Pro, and EX3Del), *PRKN* B ND40067 (F44, EX4-7DEL c203_204 DEL AG). Fibroblasts were cultured in EMEM as previously described (38). iNPCs were generated as previously described (39). iNPCs were maintained in DMEM/F12; N2, B27 supplements (17502001, 17504001; Gibco) and FGFb (100-18B; Peprotech) in fibronectin (FC010-10MG; Millipore)-coated tissue culture dishes and routinely sub-cultured every 2–3 d using accutase (A6964; Sigma-Aldrich) for detachment. Neurons were differentiated from iNPCs as previously described (40). Briefly, iNPCs were plated in a six-well plate and cultured for 2 d in DMEM/F-12 medium supplemented with 1% NEAA (nonessential amino acids Lonza, BE13-114E), 2% B27 (17504001; Invitrogen) and 2.5 $\mu$M of DAPT (2634; Tocris). On day 3, DAPT was removed, and the medium was supplemented with 1 $\mu$M SAG (smoothened agonist, 566660; Millipore) and FGF8 (75 ng/ml; Peprotech) for additional 10 d. Neurons were replated and subsequently SAG and FGF8 were withdrawn and replaced with BDNF (30 ng/ml; Peprotech 100-25), GDNF (30 ng/ml; Peprotech, 450-10), TGF-$\beta$3 (2 mM; Peprotech, 100-36e), and dcAMP (2 mM, D0627-250MG; Sigma-Aldrich) for 15 d.

## Antibodies and reagents

Antibodies and other reagents used were as follows: anti-USP30 (HPA016952, 1:500; Sigma-Aldrich), anti-TOMM20 (11802-1-AP, 1:1,000; ProteinTech), mouse anti-actin (66009-1-Ig, 1:10,000; ProteinTech), anti–phospho-Ubiquitin Ser65 (ABS1513-I, 1:1,000; Millipore), anti-SYNJ2BP (HPA000866, 1:1,000; Sigma-Aldrich), oligomycin A (75351; Sigma-Aldrich), and antimycin A (A8674; Sigma-Aldrich).

## Preparation cell lysates and Western blot analysis

Cultured cells were lysed with NP-40 (0.5% NP-40 [74385; Fluka], 25 mM Tris–HCl pH 7.5 [BP152; Fisher Bioreagents], 100 mM NaCl [S316060; Thermo Fisher Scientific], and 50 mM NaF [S7920; Sigma-Aldrich]) lysis buffer, supplemented with MPI (mammalian protease inhibitor) cocktail (P8340; Sigma-Aldrich) and Phosstop (04906837001; Roche). Proteins were resolved using SDS–PAGE (Invitrogen NuPage gel 4–12%, NP0321/NP0335), transferred to nitrocellulose membrane (10600002; Amersham), blocked in 5% milk

---

and SYNJ2BP ubiquitylation, as well as increasing levels of phospho-Ser65 Ubiquitin (pUb). Cells were treated for 4 h ± AO (1 $\mu$M) in the absence or presence of 1 $\mu$M CMPD-39. **(E, F, G)** Graphs show quantification of AO treated samples for mono-ubiquitylated TOMM20 as a percentage of total TOMM20 (F) and the pUb signal normalised to AO treated SHSY5Y cells (G) in the 38–76 kD range, for two independent experiments represented by (E) (error bars indicate the range). **(B, E)**: Black arrow indicates unmodified species, and red and blue arrow heads indicate the mono- and multi-ubiquitylated species, respectively. Source data are available for this figure.

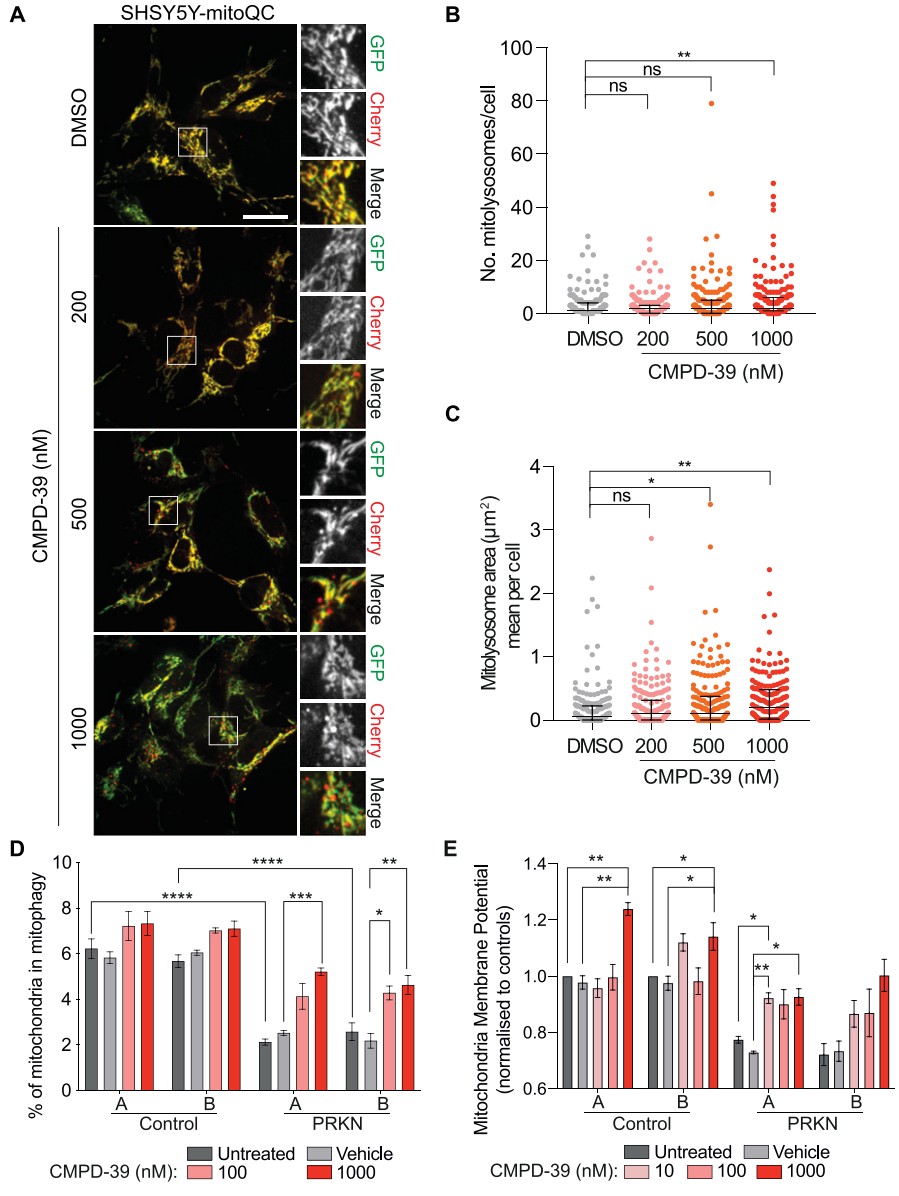

**Figure 3. Enhancement of basal mitophagy by pharmacological inhibition of USP30.**
**(A)** Representative images of SHSY5Y-mitoQC cells. Cells were treated with DMSO or CMPD-39 (200, 500, or 1,000 nM) for 96 h before imaging. Scale bar 20 μm. **(B, C)** Quantification of the data from three independent experiments is shown. Median and interquartile range are indicated; 65 cells were quantified per condition. **(B)** Graph illustrates the number of mitolysosomes. **P < 0.01, One-way ANOVA with Dunnett's multiple comparisons test. **(C)** Graph shows the mean mitolysosome area per cell. *P < 0.05, **P < 0.01, one-way ANOVA with Dunnett's multiple comparisons test.
**(D)** Mitophagy index for patient derived dopaminergic iNeurons derived from two control individuals and two individuals with PRKN loss of function mutations. CMPD-39 was administered for 24 h before measurement. Error bars indicate SEM. Two-way ANOVA with Tukey's multiple-comparisons test. *P < 0.05, **P < 0.01, ***P < 0.001, ****P < 0.0001.
**(E)** Mitochondrial membrane potential in dopaminergic neurons derived from control and PRKN patients. CMPD-39 was administered for 24 h before measurement. Error bars indicate SEM. Data are normalised to gender matched untreated control. Statistical analysis was conducted on raw non-normalised data using Mixed effect analysis with Tukey's multiple-comparisons test. *P < 0.05, **P < 0.01.
Source data are available for this figure.

(Marvel) or 5% BSA (41-10-410; First Link) in TBS (20 mM Tris–Cl, pH 7.6, and 150 mM NaCl) supplemented with Tween-20 (10485733; Thermo Fisher Scientific), and probed with primary antibodies overnight. Visualisation and quantification of Western blots were performed using IRdye 800CW (mouse 926-32212 and rabbit 926-32213), and 680LT (mouse 926-68022 and rabbit 926-68023) coupled secondary antibodies and an Odyssey infrared scanner (LI-COR Biosciences).

## Activity probe assay

Cells were mechanically homogenised in HIM buffer (200 mM mannitol; M4125; Sigma-Aldrich), 70 mM sucrose (S7903; Sigma-Aldrich), 1 mM EGTA (E4378; Sigma-Aldrich), and 10 mM HEPES-NaOH, pH 7.4 (H3375; Sigma-Aldrich) supplemented with 1 mM DTT (D11000; Melford) to obtain a PNS

(post-nuclear supernatant). Briefly, SHSY5Y cells were washed with ice-cold PBS (, 14200067; Gibco) supplemented with 1 mM DTT and then collected by scraping and centrifugation at 1,000g for 2 min. Cell pellets were washed with HIM buffer and then resuspended in HIM buffer supplemented with 1 mM DTT. Cells were mechanically disrupted by shearing through a syringe with a 27G needle, followed by three times through an 8.02 mm diameter "cell cracker" homogenizer using an 8.01 mm diameter ball bearing. The resulting homogenate was cleared from nuclei and unbroken cells by centrifugation at 600g for 10 min to obtain the PNS. Homogenates were incubated with HA-Ub-PA probe (kind gift from Yogesh Kulathu, University of Dundee) at 1:100 (w/w) for 10 min at 37°C. The reaction was stopped by the addition of sample buffer and heating at 95°C. To test drug engagement with USP30, intact cells were treated with CMPD-39 for 2 h at 37°C before homogenization followed by probe incubation.

**A**

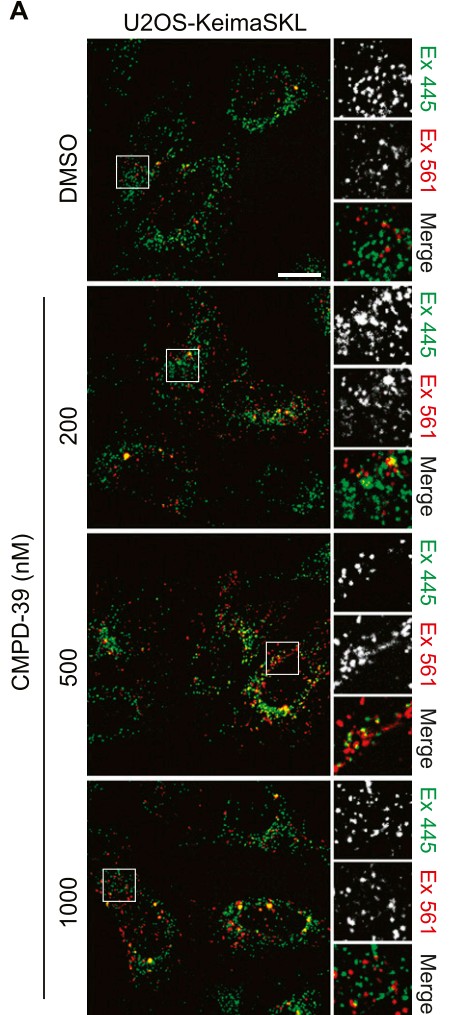

**B**

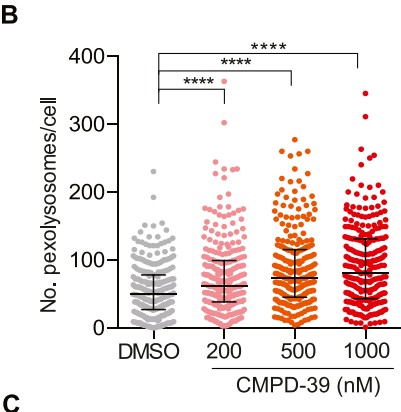

**C**

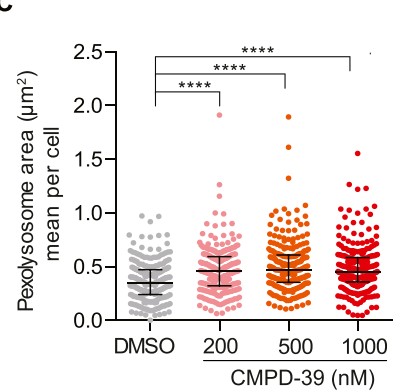

**Figure 4. Enhancement of basal pexophagy by pharmacological inhibition of USP30.**
**(A)** Representative images of U2OS-Keima-SKL cells. Cells were treated with DMSO or CMPD-39 (200, 500 or 1,000 nM) for 96 h before imaging. Scale bar 20 $\mu$m. **(B, C)** Quantification of the data is derived from three independent experiments. Median and interquartile range are indicated; 85 cells were quantified per condition. **(B)** Graph indicates the number of pexolysosomes in U2OS-Keima-SKL cells. One-way ANOVA with Dunnett's multiple comparisons test, ****$P$ < 0.0001. **(C)** Graph indicates the mean pexolysosome area per cell. One-way ANOVA with Dunnett's multiple-comparisons test, ****$P$ < 0.0001. Source data are available for this figure.

### Live cell imaging

SHSY5Y cells stably expressing mCherry-GFP-Fis1(101-152) (SHSY5Y-mitoQC) or U2OS cells stably expressing Keima-SKL (U2OS-Keima-SKL) were treated every 24 h over a 96 h time course with 200, 500 nM and 1 $\mu$M of CMPD-39. 2 × $10^5$ cells were re-plated onto an IBIDI $\mu$-Dish (81156; IBIDI) 2 d before live-cell imaging with a 3i Marianas spinning disk confocal microscope (63× oil objective, NA 1.4, Photometrics Evolve EMCCD camera, Slide Book 3i v3.0). Cells were randomly selected using the GFP (mitoQC) or Keima-445 (Keima-SKL) signal and images acquired sequentially using the following settings. SHSY5Y-mitoQC: 488 nm laser, 525/30 emission; 561 nm laser, 617/73 emission, U2OS-Keima-SKL: 445 nm laser, 525/30 emission; 561 nm laser, 617/73 emission.

To assess mitophagy in live iNeurons, a modified version of the protocol in Schwartzentruber et al (2020) was adopted (27). Briefly, cells were incubated for 1 h at 37°C with 1 $\mu$M TMRM (Tetramethylrhodamine, Methyl Ester, Perchlorate, T668; Invitrogen), 1 $\mu$M LysoTracker Green (L7526; Invitrogen) and 1 $\mu$M Hoechst dye (B2883; Sigma-Aldrich), before washing

to remove fluorescent probes. Images were captured by the Opera Phenix system (Perkin Elmer) in time lapse, every 18 min in the same fields of view, minimum eight fields of view per well and 5 z slices. Images were taken using the following settings; 488 nm laser, emission 500–550; 568 nm laser, emission 570–630; 405 nm laser, emission 435–480.

### Mitochondrial membrane potential measurements

Neurons were incubated with 80 nM TMRM and 1 $\mu$M Hoechst for 60 min at 37°C. The medium containing the dyes was replaced with MEM without phenol red (51200038; Gibco) and cells were imaged using Opera Phenix (Perkin Elmer) in live conditions (37°C, 5% $CO_2$). 20 fields of view were captured, in 6 z planes per condition, capturing a minimum of 100 cells per condition per differentiation. Data were generated from a minimum of three biological differentiations of each line. Images were taken using the following settings, 568 nm laser, emission 570–630 nm; 405 nm laser, emission 435–480 nm. Harmony (Perkin Elmer) was used for data analysis.

## Image quantification

Analysis of mitophagy and pexophagy levels in SHSY5Y-mitoQC and U2OS-Keima-SKL cells was performed using the "mito-QC Counter" implemented in FIJI v2.0 software (ImageJ, NIH) as previously described (41). For mitophagy, the following parameters were used: radius for smoothing images = 1.5, ratio threshold = 0.8, and red channel threshold = mean. For pexophagy, the following parameters were used: radius for smoothing images = 1.5, ratio threshold = 1.9, and red channel threshold = mean. Mitophagy and pexophagy analysis was performed for three independent experiments with 65 and 85 cells per condition, respectively. Images generated from the live iNeuron imaging experiments were analyzed using Harmony (Perkin Elmer software). We developed protocols to segment nucleus, image regions containing cytoplasm, mitochondria, lysosomes, and autolysosomes containing mitochondria. Maximal projection images were used for analysis. Mitochondria contained within lysosomes segmentation was set up in such a way to identify a mitophagy event when the overlap between mitochondria and lysosome was 100%.

## Statistical analysis

$P$-values are indicated as *$P < 0.05$, **$P < 0.01$, ***$P < 0.001$ and ****$P < 0.0001$ and derived either by one-way ANOVA with a Dunnett's multiple comparisons post hoc test (Figs 3B and C and 4B and C), or a two-way ANOVA with Tukey's multiple comparisons (Fig 3D) or mixed-effects analysis with Tukey's multiple comparisons (Fig 3E) using GraphPad Prism.

# Supplementary Information

# Acknowledgements

hTERT-RPE1-YFP-PRKN cells and SHSY5Y-mitoQC cells were a kind gift of Jon Lane (University of Bristol) and Ian Ganley (University of Dundee) respectively. This study was funded by Celgene and Bristol Myers Squibb. We thank Richard Hargreaves and Jeff Schkeryantz for their encouragement and enablement of this work. FG Barone is a Wellcome Trust funded PhD student (102172/B/13/Z).

## Author Contributions

EV Rusilowicz-Jones: conceptualization, data curation, formal analysis, investigation, funding acquisition, methodology, and writing—original draft, review, and editing.
FG Barone: conceptualization, data curation, formal analysis, investigation, and writing—review and editing.
F Martins Lopes: formal analysis and investigation.
E Stephen: formal analysis and investigation.
H Mortiboys: conceptualization, supervision, funding acquisition, project administration, and writing—review and editing.
S Urbe: conceptualization, supervision, funding acquisition, visualization, project administration, and writing—original draft, review, and editing.
MJ Clague: conceptualization, supervision, funding acquisition, project administration, and writing—original draft, review, and editing.

## Conflict of Interest Statement

The authors declare that they have no conflict of interest.

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
