## [Reviewer comments · Life Science Alliance]

Benchmarking a highly selective USP30 inhibitor for enhancement of mitophagy and pexophagy.

Emma Rusilowicz-Jones, Francesco Barone, Fernanda Lopes, Elezabeth Stephen, Heather Mortiboys, Sylvie Urbe, and Michael Clague

DOI: <https://doi.org/10.26508/lsa.202101287>

Corresponding author(s): Michael Clague, University of Liverpool

Review Timeline:	Submission Date:	2021-11-03
	Editorial Decision:	2021-11-05
	Revision Received:	2021-11-10
	Accepted:	2021-11-12

Transaction Report:

Please note that the manuscript was previously reviewed at another journal and the reports were taken into account in the decision-making process at *Life Science Alliance*. Since the original reviews are not subject to Life Science Alliance's transparent review process policy, the reports and author response cannot be published.

November 5, 2021

RE: Life Science Alliance Manuscript #LSA-2021-01287

Prof. Michael J. Clague
University of Liverpool
Cellular and Molecular Physiology, Biomedical Sciences
Crown St.
Liverpool, Merseyside L69 3BX
United Kingdom

Dear Dr. Clague,

Thank you for submitting your revised manuscript entitled "Benchmarking a highly selective USP30 inhibitor for enhancement of mitophagy and pexophagy.". We would be happy to publish your paper in Life Science Alliance pending final revisions necessary to meet our formatting guidelines.

-please consult our manuscript preparation guidelines <https://www.life-science-alliance.org/manuscript-prep> and make sure your manuscript sections are in the correct order.

-please add an Author Contributions section to your main manuscript text

-please add a Conflict of Interest statement to your main manuscript text

-please upload your main and supplementary figures as single files

-please upload your main manuscript text as an editable doc file

-please add the twitter handle of your host institute/organization as well as your own or/and one of the authors in our system

A. FINAL FILES:

B. MANUSCRIPT ORGANIZATION AND FORMATTING:

Sincerely,

November 12, 2021

RE: Life Science Alliance Manuscript #LSA-2021-01287R

Prof. Michael J. Clague
University of Liverpool
Cellular and Molecular Physiology, Biomedical Sciences
Crown St.
Liverpool, Merseyside L69 3BX
United Kingdom

Dear Dr. Clague,

Thank you for submitting your Research Article entitled "Benchmarking a highly selective USP30 inhibitor for enhancement of mitophagy and pexophagy.". It is a pleasure to let you know that your manuscript is now accepted for publication in Life Science Alliance. Congratulations on this interesting work.

DISTRIBUTION OF MATERIALS:

Again, congratulations on a very nice paper. I hope you found the review process to be constructive and are pleased with how the manuscript was handled editorially. We look forward to future exciting submissions from your lab.

Sincerely,
